# The Risk of Reinfection or Primary Hepatitis E Virus Infection at a Liver Transplant Center in Brazil: An Observational Cohort Study

**DOI:** 10.3390/v16020301

**Published:** 2024-02-16

**Authors:** Michelle Zicker, João R. R. Pinho, Eliane A. R. Welter, Bianca D. Guardia, Paulo G. T. M. da Silva, Leonardo B. da Silveira, Luís F. A. Camargo

**Affiliations:** 1Division of Infectious Diseases, Department of Internal Medicine, Universidade Federal de São Paulo, São Paulo 04023-062, Brazil; luisfacamargo@uol.com.br; 2Research and Development Sector, Clinical Laboratory, Hospital Israelita Albert Einstein, São Paulo 05652-900, Brazil; 3Liver Transplant Program, Hospital Israelita Albert Einstein, São Paulo 05652-900, Brazil; 4Faculdade Israelita de Ciências da Saúde Albert Einstein, São Paulo 05653-120, Brazil

**Keywords:** hepatitis E, liver transplantation, prevalence, incidence, natural history, Brazil

## Abstract

The hepatitis E virus is a major etiological agent of chronic hepatitis in immunosuppressed individuals. Seroprevalence in the liver transplantation setting varies according to the seroprevalence of the general population in different countries. This was a prospective cohort study of liver transplant recipients in southeastern Brazil. Recipients were systematically followed for one year, with the objective of determining the prevalence, incidence, and natural history of HEV infection in this population. We included 107 liver transplant recipients and 83 deceased donors. Positivity for anti-HEV IgG was detected in 10.2% of the recipients and in 9.7% of the donors. None of the patients tested positive for HEV RNA at baseline or during follow-up. There were no episodes of reactivation or seroconversion, even in cases of serological donor-recipient mismatch or in recipients with acute hepatitis. Acute and chronic HEV infections seem to be rare events in the region studied. That could be attributable to social, economic, and environmental factors. Our data indicate that, among liver transplant recipients, hepatitis E should be investigated only when there are elevated levels of transaminases with no defined cause, as part of the differential diagnosis of seronegative hepatitis after transplantation.

## 1. Introduction

As has been demonstrated previously, the hepatitis E virus (HEV) is a major etiological agent of acute hepatitis and liver failure in the general population. It has recently been shown to be a cause of chronic hepatitis and cirrhosis in immunosuppressed individuals, especially in solid organ transplant (SOT) recipients [1,2,3,4,5].

Hepatitis E is a non-enveloped RNA virus of the family *Hepeviridae*, which has two subfamilies, *Parahepevirinae* and *Orthohepevirinae* [6]. Members of the *Orthohepevirinae* infect mammals and birds and are further classified into four genera: *Paslahepevirus*, *Avihepevirus*, *Rocahepevirus*, and *Chirohepevirus* [6]. The genus *Paslahepevirus* includes the species *P. balayani*, which infects humans and some mammalian species [7]. Four major HEV genotypes (HEV-1 to HEV-4) within the species *Paslahepevirus balayani* infect humans. Genotype 3 (HEV-3), which is transmitted mainly through the consumption of pork, is the main genotype responsible for chronic hepatitis in immunosuppressed individuals, whereas genotypes 1, 4, and 7 have been implicated only in rare cases [8].

The HEV RNA genome contains three *Open Reading Frames* (ORF): ORF1 encodes a polyprotein that includes an RNA dependent RNA polymerase; ORF2 encodes the capsid protein; and ORF3 encodes a protein involved in viral egress. ORF 4, recently described for HEV-1, has a supporting role for viral polymerase [9].

In patients with acute HEV infection, peak viremia occurs during the incubation period and early phase of disease, but HEV RNA does not persist for long and becomes undetectable in the blood about 3 weeks after the onset of symptoms. The window of detectable RNA is therefore narrow. On the other hand, viral shedding lasts approximately 4–6 weeks in the stool [10,11]. As an RNA genome virus, HEV is not expected to persist in latency in hepatocytes, although few studies have addressed reactivation in the context of immunosuppression [12,13].

Most of the data on hepatitis E come from prevalence studies conducted in high-income countries, mainly in Europe, and seroprevalence rates in liver transplant recipients (LTRs) vary according to the seroprevalence in the general population [5]. Data regarding the clinical evolution of hepatitis E in SOT recipients are discordant, with reports ranging from benign evolution to a high frequency of chronic disease [14,15]. Given that geographic, socioeconomic, and cultural differences seem to lead to variability in the presentation of HEV infection, the aim of this study was to determine the prevalence, incidence, genotype, and natural history of HEV infection among LTRs in southeastern Brazil.

## 2. Materials and Methods

### 2.1. Study Design and Patient Selection

This was a prospective study of LTRs submitted to the periodic collection of blood samples to identify markers of HEV infection. We included patients ≥ 18 years of age who underwent liver transplantation between June 2019 and December 2020 at the organ transplant center of the Hospital Municipal Vila Santa Catarina, in the city of São Paulo, Brazil. The hospital is operated by the Hospital Israelita Albert Einstein, also based in the city of São Paulo.

Recipients were followed for one year, during which time blood samples were collected periodically for serology (for anti-HEV IgG and IgM antibodies) and for quantitative reverse transcription PCR (qRT-PCR): in the pretransplant period; monthly for the first 6 months after transplantation; and at 9 and 12 months after transplantation. That periodicity was necessary to avoid missing transient viremia and to determine the behavior of the viral load over time. When a qRT-PCR was positive for HEV, genotyping was performed. When available, blood samples collected from subsequently deceased donors were also submitted to serology and qRT-PCR. When an LTR developed acute hepatitis during the follow-up period, we performed additional serology for anti-HEV IgM and IgG antibodies, as well as qRT-PCR for HEV, to determine the incidence of acute hepatitis E in the posttransplant period. In recipients found to be infected with HEV, qRT-PCR was also performed during clinical events that required hospitalization, such as any other severe infection or acute rejection. Epidemiological, clinical, and biochemical data were collected from all of the recipients evaluated.

Blood samples were collected at Hospital Municipal Vila Santa Catarina and sent to the Clinical Laboratory at Hospital Israelita Albert Einstein. Samples were collected in S-Monovette^®^ Serum Gel Lightprotect for HEV serology and S-Monovette^®^ EDTA Gel K2, 7.5 mL for HEV PCR (SARSTEDT AG & Co. KG, Nümbrecht, Germany) and sent to the Central Laboratory for centrifugation at 2500× *g* for 10 min in a maximal interval time of two hours. Serum or plasma were separated in 1.0 mL aliquots and stored in 1.5 mL tubes at −80 °C in a Thermo -80 Freezer Revco EXF (Thermo Fisher Scientific, Marietta, OH, USA). This laboratory is accredited by the College of American Pathology and by the Brazilian Society of Clinical Pathology. All reactions follow strict quality control procedures, including the obligatory use of positive and negative controls in each run beyond the ones furnished by the kit manufacturers.

For HEV tests, positive controls were obtained from previous studies carried out by our group:Serological positive samples were available from a previous study carried out with HIV-infected patients [16].HEV RNA positive samples were available from a previous study carried out with pigs. These pig viruses were characterized as subtypes 3c and 3f, which have also been found in human samples [17].To be sure that our technique was good to detect HEV in humans, we have ordered the “1st World Health Organization International Standard for Hepatitis E Virus RNA Nucleic Acid Amplification Techniques (NAT)-Based Assays”, a genotype 3a strain of HEV, derived from the plasma of a blood donor [18] and also genotypes 3c and 3f strains bought at that time from Qnostics Ltd. (Glasgow, Scotland, UK). After all the protocols proposed by the College of American Pathology, the technique was implemented in the routine with an analytical sensitivity of 50 UI/mL and an accuracy of 100% using the available positive and negative samples.For the first assays, the positive controls were those described above, and up to the moment we obtained a very strong positive sample that was also submitted to a full-length sequence and confirmed as HEV genotype 3f.

### 2.2. Definitions

To characterize a new or reactivated HEV infection, the following definitions were used:HEV infection: anti-HEV IgM positivity, anti-HEV IgG positivity, or positivity for HEV on qRT-PCR.Primary HEV infection: anti-HEV IgM positivity, anti-HEV IgG positivity, or positivity for HEV on qRT-PCR in a patient previously testing negative for anti-HEV IgG or IgM antibodies.Chronic HEV infection: positivity for HEV on qRT-PCR for a period of three months or more.Acute hepatitis: increase to at least double the normal value in the concentrations of the hepatic transaminases alanine aminotransferase (ALT) and aspartate aminotransferase or of the canalicular enzymes gamma-glutamyl transferase and alkaline phosphatase.

### 2.3. Serology

In serum samples, we quantified anti-HEV IgM and IgG antibody titers, using commercial in vitro kits (recomWell HEV IgM/IgG; Mikrogen GmbH, Neuried, Germany), which are based on the principle of an indirect sandwich ELISA with recombinant antigens from the open reading frame 2 and open reading frame 3 regions of the HEV genome, expressed in *Escherichia coli*.

### 2.4. HEV RNA Detection

All samples were submitted for molecular analysis. For the extraction of viral RNA, we used the QIAamp MinElute Virus Spin extraction kit (QIAGEN, Hilden, Germany), and we used the RealStar HEV RT-PCR kit 2.0 (Altona Diagnostics, Hamburg, Germany) for the detection of HEV. The assay includes a heterologous amplification system (internal control) to identify possible RT-PCR inhibition and to confirm the integrity of the reagents in the kit.

To rule out the possibility of false-negative results, we also subjected every sample to a laboratory-developed test (LDT) that was designed as a pan genotypic assay for the detection and quantification of HEV RNA, including all genotypes belonging to the Orthohepevirus subfamily. The primers and probe covered the conserved ORF3 region, amplifying a 70 bp sequence with the following sequences: forward primer, 5′-RGTRGTTTCTGGGGTGAC-3′; reverse primer, 5′-AKGGRTTGGTTGGRTGA-3′; probe, 5′-FAM-TGAYTCYCARCCCTTCGC-TAMRA-3′. Samples were submitted to the following protocol: reverse transcription (50 °C, 30 min); initial denaturation (95 °C, 15 min); followed by 45 cycles of denaturation (95 °C, 10 s); annealing (51 °C, 30 s); and extension (60 °C, 20 s) [19].

### 2.5. Statistical Analysis

Means and standard deviations were reported for continuous variables. For categorical variables, frequencies and percentages were reported. Statistical analyses were performed with JASP software, version 0.141.0 [20].

### 2.6. Ethical Aspects

All procedures were performed in accordance with the Ethical Principles for Medical Research Involving Human Subjects outlined in the 2013 Declaration of Helsinki and with the guidelines established in the 2018 Declaration of Istanbul. The study was approved by the Research Ethics Committees of the Hospital Israelita Albert Einstein and the Federal University of São Paulo. All subjects gave their informed consent for inclusion before they participated in the study.

## 3. Results

### 3.1. HEV Infection in the Pretransplant Period

Between June 2019 and December 2020, we recruited 107 LTRs and 83 donors. We collected a total of 765 blood samples.

In the pretransplant period, the anti-HEV IgG test result was positive in 11 (10.2%) of the recipients, positive in eight (9.7%) of the donors, and inconclusive in three (1.6%) of the recipients. The anti-HEV IgM test result was positive in one (0.9%) of the recipients and inconclusive in one (1.2%) of the donors.

The demographic, clinical, and epidemiological characteristics of the recipients are described in Table 1. Among the 107 recipients, the mean age was 53.5 years, and most (72.9%) were men. Almost all (96.3%) of the recipients lived in urban areas, and the majority (83.2%) reported eating pork.

### 3.2. Positive Anti-HEV IgG Receptors in Pretransplant

Among the 11 recipients who were anti-HEV IgG-positive in the pretransplant period, there were no cases of reinfection, as would have been identified through the detection of HEV RNA in the follow-up examinations, even among the four recipients who were diagnosed with acute hepatitis. Among the four recipients with acute hepatitis, two had an episode of acute cellular rejection, one developed stenosis of the biliary anastomosis, and one had metabolic acidosis with hyperkalemia, with no apparent hepatic cause.

There was no significant increase in anti-HEV IgG titers in the follow-up examinations. Four of the 11 recipients who were anti-HEV IgG-positive in the pretransplant period had an anti-HEV IgG titer < 20 U/mL in at least one follow-up examination. The seven remaining recipients maintained an anti-HEV IgG titer > 25 U/mL throughout the one-year follow-up period.

### 3.3. Anti-HEV IgG-Positive Donors

Eight recipients, six of whom were anti-HEV IgG-negative in the pretransplant period, received organs from anti-HEV IgG-positive donors. Among those eight recipients, no seroconversions were identified; nor was any HEV RNA detected in the follow-up examinations.

### 3.4. HEV RNA

In the pretransplant setting and during the follow-up period, no HEV RNA was detected by either the RealStar HEV RT-PCR kit 2.0 or the LDT PCR assay.

### 3.5. Comparison between Anti-HEV IgG Positive and Negative Recipients

The baseline characteristics and outcomes of the pretransplant anti-HEV IgG positive and anti-HEV IgG negative recipients are shown in Table 2. There was no statistically significant difference between the two groups.

## 4. Discussion

The prevalence of anti-HEV IgG antibodies in the context of liver transplantation is highly variable between countries and even between regions within the same country. A recent meta-analysis that included 18 studies from the United States, Thailand, China, and various European countries, found the pooled estimated prevalence of anti-HEV IgG positivity among LTRs to be 27.2% [21]. The differences in HEV prevalence may be a reflection of geographic and socioeconomic diversity, as well as differences in lifestyle habits and in the performance of the serology assay employed. In France, HEV-3 infection is quite common, especially in the southwestern region of the country, where the seroprevalence of HEV-3 among SOT recipients is nearly 40% [22]. The high prevalence in that region is probably related to the higher consumption of raw or undercooked pork. In other countries, the prevalence rates for HEV-3 infection are lower, such as the 2.9% and 7.1% reported for Japan and Argentina, respectively [23,24]. It has been extensively reported that the Wantai HEV IgG ELISA has the highest sensitivity for HEV detection, with excellent specificity as well. The Mikrogen kit, which was employed in the present study, has been reported to have a sensitivity of 96.6%, a specificity of 97.1%, and good performance in detecting anti-HEV-IgG, according to two meta-analyses that compared various commercial kits [2,25].

The anti-HEV IgG prevalence of 10.2% found in the present study is comparable to those reported in two studies employing the Mikrogen kit in the evaluation of LTRs in southeastern Brazil (8.1% and 8.2%), albeit lower than the 18.7% reported in a study employing the Wantai kit in the evaluation of LTRs in the southern region of the country [26,27,28]. Southern is the region with the highest number of pig farms as well as the highest consumption of pork, and the prevalence of anti-HEV IgG positivity has been reported to be higher (up to 40.3%) there than in any other region of the country [28,29,30]. Brazil is also the fifth largest country in the world and has extremely diverse socioeconomic and sanitation conditions, which influence the dynamics of infectious diseases. The fact that a large majority of the recipients in our study were residents of the southeastern region could explain the fact that the prevalence of anti-HEV antibody positivity in our sample was lower than that reported in studies carried out in the southern region.

A systematic review and meta-analysis estimated that the overall seroprevalence of HEV infection in the adult population in Brazil was 6%. In a subgroup analysis, the prevalence in blood donors was 7.0%, whereas in the general population it was 3% [31]. Anti-HEV IgG seroprevalence was found to be 4.3% and 9.8% in two studies conducted in blood donors in the southeastern part of the country [32,33]. In our study, the prevalence of anti-HEV IgG positivity among the recipients was similar to that observed among the donors, suggesting that chronic liver disease is not a risk factor for HEV infection. Similarly, the prevalence of anti-HEV IgG positivity was found to be comparable between blood donors and LTRs in the southern region of Brazil [28].

In the present study, no HEV RNA was detected during the follow-up of the 11 recipients who were anti-HEV IgG-positive before transplantation. Therefore, qRT-PCR revealed no reinfection, even among the recipients with acute hepatitis. One male recipient who, in the pretransplant period, tested positive for anti-HEV IgG and negative for anti-HEV IgM, subsequently tested positive for anti-HEV IgM, with an elevated IgG titer (>125 U/mL). However, he also tested positive for cytomegalovirus, with a high viral load, and it was not possible to confirm a new HEV infection. The antibody concentration required to protect against HEV infection has not been well established. In a rhesus monkey model of HEV infection, previous HEV infection was shown to be capable of conferring cross-protection against different genotypes, even though HEV RNA was detected in animals with reinfection [34]. A study that evaluated reinfection suggested that antibody levels lower than 7 U/mL are not protective [34].

During the follow-up examinations of the anti-HEV IgG-negative recipients, no HEV RNA was detected, and there was no seroconversion, even in cases of serological donor-recipient mismatch or in recipients with acute hepatitis. Thus, neither a primary nor a chronic infection was diagnosed. Certain factors, like the rate of HEV RNA detection within and outside the context of SOT, the sensitivity of the commercial kit used, and aspects of the immune response of the recipient, must be considered when analyzing this finding.

The HEV RNA detection rate reported in cross-sectional and prospective studies of LTRs is low, ranging from 0% to 4% in most published articles [3,13,15,22,23,24,35,36,37,38,39,40,41,42,43,44,45,46,47,48,49,50,51,52]. However, in a prospective study conducted in Thailand, involving 711 samples collected from LTRs who were followed for 12 months, HEV RNA was detected in 7.7% of the 91 recipients, among whom the authors found the prevalence of anti-HEV IgG positivity to be 52.7% [48]. In Brazil, within and outside the context of SOT, there is a low detection rate for HEV RNA, which has been reported to be undetectable in some studies. In a study of 294 LTRs in southeastern Brazil, HEV RNA was detected in only 5.8% [27]. In a study of 80 LTRs in the southern region of the country, no HEV RNA was detected in any of the recipients [28]. Outside the context of SOT, two studies, both conducted in southeastern Brazil, are notable. In a study of 354 HIV-infected patients, among whom the prevalence of anti-HEV IgG positivity was 10.7%, no HEV RNA was detected. In another, more recent study, which included 400 patients with elevated transaminases, an HEV RNA detection rate of 4% was reported [16,53].

The kit employed for qRT-PCR in the present study has been used in studies of SOT in Brazil and Europe and has good analytical sensitivity [28,40,54]. To avoid false-negative results, the samples evaluated in the present study were submitted to a pangenotypic PCR-based assay, as previously described, and there was no HEV RNA detection as well [19].

One group of authors suggested that HEV RNA would be detected almost exclusively in the setting of chronic infection, which could be related to the fact that HEV infection typically evolves to chronicity in immunosuppressed patients or to the short window for HEV RNA detection in the course of self-limited acute infection [40]. The effect of the latter factor can be mitigated by collecting blood samples more frequently.

Transmission of HEV via graft is a rare event in the SOT setting. There has been at least one reported case of a LTR who was infected via the donated organ, subsequently progressing to chronicity and liver cirrhosis [55]. There has also been a report of two kidney transplant recipients who developed chronic HEV infection after receiving organs from the same donor [56]. The fact that HEV transmission via graft is rare is likely explained by the relatively rapid viral clearance in immunocompetent patients and the rarity of latent viruses in donated organs. Primary HEV infection from environmental exposure after transplantation has been reported more frequently, with the highest rates reported for France and India, whereas rates below 2% have been reported for other countries [13,15]. That discrepancy could be attributed to the way in which we preselected individuals for transplantation (taking into account social and environmental conditions), less exposure to risk factors associated with HEV infection (due to the posttransplant care recommended), the probable predominance of HEV-3 in Brazil, and the higher proportion of patients who were residents of the southeastern region, where basic sanitation is better and the rates of pork consumption are lower than in other regions of the country.

In a study conducted recently in the United States, HEV RNA was detected in only one of 203 LTRs, suggesting that HEV chronicity is rare within the liver transplant population in that country [52]. It is possible that HEV chronicity is also rare in Brazil and that larger studies conducted in different regions of the country would lead to the identification of acute infection after transplantation, with occasional conversion to chronic infection and consequent detection of HEV RNA. The low seroprevalence rate among the recipients evaluated in the present study might have also contributed to the failure to identify HEV RNA in the samples analyzed.

As there was no HEV RNA detected, it was not possible to determine the HEV genotype in the infected patients. Molecular characterization of the HEV strains detected in humans and pigs in Brazil showed that all belonged to HEV-3 [30,57,58,59,60,61]. Further characterization of human strains identified subtypes 3b and 3i. Despite the low sanitary conditions of some regions, HEV-1 and HEV-2 were not detected in Brazil, suggesting that Brazilian epidemiology may be similar to that of industrialized countries [30].

The originality of our study lies in the longitudinal follow-up of a large number of patients, with the systematic collection of blood samples to assess HEV infection status rather than the point prevalence of infection and disease caused by the virus. Our study has some limitations. The acute viremic cases of hepatitis E described in Brazil are rare, although many descriptions of positive serologic cases have already been reported throughout the country. It is possible that the infection is more common than it was described here, but we would have to have more frequent sampling to determine it and must perform a multicenter study to get a better analysis of the impact of this disease in the country. The fact that it was conducted at a single center could have created a selection bias. In addition, the follow-up period was relatively short (only one year). Furthermore, the sample was fairly homogeneous in terms of regional representation, given that most of the patients evaluated resided in southeastern Brazil.

We can conclude that, in the scenario of liver transplantation in Brazil, hepatitis E should be investigated only in recipients with elevated levels of transaminases or canalicular enzymes with no defined cause, as part of the differential diagnosis of seronegative hepatitis after transplantation. Therefore, we believe that pretransplant screening and systematic testing for HEV infection are unnecessary, as is the case for infection with cytomegalovirus and certain other pathogens. Given the size and heterogeneity of the population of Brazil, in terms of habits and socioeconomic conditions, we believe that there is a need for additional studies to evaluate LTRs in other regions of the country. With more extensive data on the prevalence and incidence of HEV infection in the country, it would be possible to determine the cost-effectiveness of systematic screening for such infection in the setting of liver transplantation in Brazil.

## Figures and Tables

**Table 1 viruses-16-00301-t001:** Baseline characteristics of liver transplant recipients evaluated between 2019 and 2020.

Characteristic	(N = 107)
Age (years), mean ± SD	53.5 ± 13.13
Male, n (%)	78 (72.9)
More than one underlying disease, n (%)	44 (41.1)
Underlying disease, n (%)	
Alcoholic liver cirrhosis	34 (31.8)
Chronic hepatitis C	24 (22.4)
Cryptogenic cirrhosis	15 (14.0)
Autoimmune hepatitis	9 (8.4)
Non-alcoholic steatohepatitis	7 (6.5)
Other	18 (16.8)
Region of birth, n (%)	
Southeastern Brazil	77 (72.0)
Northeastern Brazil	23 (21.5)
Central-west Brazil	3 (2.8)
Southern Brazil	2 (1.9)
Northern Brazil	1 (0.9)
Region of residence, n (%)	
Southeastern Brazil	101 (94.4)
Central-west Brazil	4 (3.7)
Northeastern Brazil	2 (1.9)
Resident of an urban area, n (%)	103 (96.3)
Level of education, n (%)	
None	1 (1.0)
<9 years of schooling	18 (17.1)
9 years of schooling	11 (10.4)
High school, incomplete	5 (4.8)
High school, complete	36 (34.3)
College, incomplete	12 (11.4)
College, complete	21 (20.0)
Postgraduate work	1 (1.0)
Previous blood transfusion, n (%)	56 (52.3)
Consumption of pork, n (%)	89 (83.2)
Consumption of game meat, n (%)	40 (37.4)
Consumption of seafood, n (%)	79 (73.8)
Home with sewage system, n (%)	105 (98.1)
Home with treated water, n (%)	105 (98.1)
Contact with domestic animals, n (%)	56 (52.3)
Contact with natural waters, n (%)	55 (51.4)
Previous contact with an HEV-infected individual, n (%)	2 (1.9)
Travel in the last year, n (%)	50 (46.7)
Anti-HEV IgG-positive, n (%)	11 (10.3)
Anti-HEV IgM-positive, n (%)	1 (0.9)

**Table 2 viruses-16-00301-t002:** Baseline characteristics of the pretransplant anti-HEV IgG positive (+) and anti-HEV IgG negative (−) recipients.

Characteristic	HEV IgG + (n = 11)		HEV IgG − (n = 96)		*p*-Value
**Male, n (%)**	10	(91%)	68	(71%)	0.156
**Age (years), mean ± SD**	57	(13.5)	53	(7.9)	0.338
**Region of birth, n (%)**					
Southeastern Brazil	6	(55%)	71	(75%)	0.230
Northeastern Brazil	4	(37%)	19	(20%)	
Central-west Brazil	0	(0%)	3	(3%)	
Southern Brazil	1	(8%)	1	(1%)	
Northern Brazil	0	(0%)	1	(1%)	
**Region of residence, n (%)**					
Southeastern Brazil	10	(91%)	91	(95%)	0.143
Central-west Brazil	0	(0%)	4	(4%)	
Southern Brazil	1	(9%)	1	(1%)	
**Resident of an urban area, n (%)**	11	(100%)	92	(96%)	1.0
**>9 years of schooling**	6	(45%)	69	(73%)	0.190
**Underlying disease, n (%)**					
Chronic hepatitis C	1	(9%)	23	(24%)	0.182
Alcoholic liver cirrhosis	6	(55%)	21	(22%)	
Cryptogenic cirrhosis	1	(9%)	14	(15%)	
Non-alcoholic steatohepatitis	0	(0%)	7	(7%)	
Other	3	(27%)	31	(32%)	
**More than one underlying disease, n (%)**	4	(46%)	40	(42%)	0.735
**Previous blood transfusion, n (%)**	5	(45%)	51	(54%)	0.580
**Consumption of pork, n (%)**	9	(82%)	80	(85%)	0.774
**Consumption of game meat, n (%)**	6	(55%)	34	(36%)	0.235
**Consumption of seafood, n (%)**	7	(64%)	72	(77%)	0.346
**Home with sewage system, n (%)**	11	(100%)	94	(99%)	1.0
**Home with treated water, n (%)**	11	(100%)	94	(99%)	1.0
**Contact with domestic animals, n (%)**	3	(27%)	53	(57%)	0.107
**Contact with natural waters, n (%)**	5	(45%)	50	(54%)	0.752
**Previous contact with an HEV-infected individual, n (%)**	0	(0%)	2	(2%)	1.0
**Travel in the last year, n (%)**	3	(27%)	47	(51%)	0.144
**Retransplant**	0	(0%)	5	(5%)	1.0
**1-year survival**	10	(91%)	86	(90%)	0.891

## Data Availability

The data analyzed during the current study is not publicly available, but it is available upon request from the corresponding author.

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
