# Peer review of "The Risk of Reinfection or Primary Hepatitis E Virus Infection at a Liver Transplant Center in Brazil: An Observational Cohort Study"

_viruses, 2024, doi:10.3390/v16020301_

Round 1
Reviewer 1 Report
Comments and Suggestions for Authors
Zicker et al. have investigated HEV seroprevalence among 107 liver transplantation candidates and the one-year HEV infection incidence after LT in a single transplant center of Southeastern Brazil. Methods are straightforward, with periodic blood sampling during a one-year follow-up and the use of appropriate serological and molecular tools. Liver donors’ HEV status was also investigated. A baseline seroprevalence of 10.2% was observed among LT recipients. No active infection was detected.
Globally, the manuscript must be improved.
the main message seems to be that HEV seroprevalence among LT recipients is similar to that already found for this population in the same region, but you'll have to look hard to find these data in the discussion, which is too long and should focus on the main messages.
What is the seroprevalence in southeastern Brazil in the general population? In blood donors? Most publications say the same thing: LT candidates are no more exposed than the general population. Thus, sentences found in the abstract “studies …have produced mixed results” or in introduction “have produced quite variable results” are not relevant, as seroprevalence results depend on local epidemiology, Table 2 is dispensable.
The second message is that, over the period studied, there were no acute infections in this population. There is confusion, right from the title and throughout the manuscript, between reactivation and reinfection. To my knowledge, there is no latent form of HEV, an RNA virus with strictly cytoplasmic replication, so there can be no reactivation, only reinfections. This is consistent with the fact that HEV IgG positive donors do not transmit HEV infection.
Table 1 describes in detail the whole population characteristics; we would like to see a comparison between positive and negative patients. Are there significant differences between the 2 populations?
In the introduction, the paragraph on taxonomy needs updating. It could already include certain virological notions such as the fact that HEV is an RNA+ virus with no latency, and perhaps the duration of viremia during the acute phase
In the abstract reference is made to HEV genotyping, this is dispensable. However, genotypes circulating in Brazil are not discussed, although data are available.
Reviewer 2 Report
Comments and Suggestions for Authors
The authors of this article aimed to determine the prevalence, incidence, genotype and natural history of HEV among LTRs. This article is relevant for the field. However, I do suggest certain things, which need attention, improvement and clarification to support and strengthen the overall impact of the article.
Major points for attention:
Abstract: Line 22: please remove the number 11 because the percentage is indicated.
Introduction:
For the classification please cite the latest classification released (International Committee on the Taxonomy of Viruses, ICTV.)
Methodology:
Authors should indicate the storing conditions of collected samples (indicate whether they were processed immediately after preparation or stored). Also, it should be indicated how the samples were prepared for ELISA and RNA isolation (were the RNA isolates stored before qRT-PCR?).
Authors didn’t mention whether the positive and negative controls were used for qRT-PCR (if they were used, they should be described in detail).
Also, authors should describe how did they monitor the quality of RNA isolates. Was the exogenous Internal Positive Control (IPC) added to each sample to supervise the appearance of potential PCR inhibitors?
In-house PCR based assay should be described in more details.
Computer software, search tools and databases should be cited in the reference list.
Results:
Lines 134-144 are more suitable for the chapter Discussion.
Authors didn’t mention the results of the in-house PCR based assay.
Discussion:
It would be good if the authors could discuss the limitations of the presented study.
